# Ecology and Machine Learning-Based Classification Models of Gut Microbiota and Inflammatory Markers May Evaluate the Effects of Probiotic Supplementation in Patients Recently Recovered from COVID-19

**DOI:** 10.3390/ijms24076623

**Published:** 2023-04-01

**Authors:** Lucrezia Laterza, Lorenza Putignani, Carlo Romano Settanni, Valentina Petito, Simone Varca, Flavio De Maio, Gabriele Macari, Valerio Guarrasi, Elisa Gremese, Barbara Tolusso, Giulia Wlderk, Maria Antonia Pirro, Caterina Fanali, Franco Scaldaferri, Laura Turchini, Valeria Amatucci, Maurizio Sanguinetti, Antonio Gasbarrini

**Affiliations:** 1CeMAD, Digestive Disease Center, Dipartimento di Scienze Mediche e Chirurgiche, Fondazione Policlinico Universitario “A. Gemelli” IRCCS, 00168 Rome, Italy; 2Department of Diagnostics and Laboratory Medicine, Unit of Microbiology and Diagnostic Immunology, Unit of Microbiomics and Immunology, Rheumatology and Infectious Diseases Research Area, Unit of Human Microbiome, Bambino Gesù Children’s Hospital, IRCCS, 00165 Rome, Italy; 3Laboratorio di Microbiologia Clinica, Dipartimento di Scienze di Laboratorio ed Infettivologiche, Fondazione Policlinico Universitario “A. Gemelli” IRCCS, 00168 Rome, Italy; 4GenomeUp SRL, 00144 Rome, Italy; 5Immunology Facility, Gstep, Fondazione Policlinico Universitario “A. Gemelli” IRCCS, 00168 Rome, Italy

**Keywords:** post-COVID-19, probiotic supplementation, gut microbiota

## Abstract

Gut microbiota (GM) modulation can be investigated as possible solution to enhance recovery after COVID-19. An open-label, single-center, single-arm, pilot, interventional study was performed by enrolling twenty patients recently recovered from COVID-19 to investigate the role of a mixed probiotic, containing Lactobacilli, Bifidobacteria and *Streptococcus thermophilus*, on gastrointestinal symptoms, local and systemic inflammation, intestinal barrier integrity and GM profile. Gastrointestinal Symptom Rating Scale, cytokines, inflammatory, gut permeability, and integrity markers were evaluated before (T_0_) and after 8 weeks (T_1_) of probiotic supplementation. GM profiling was based on 16S-rRNA targeted-metagenomics and QIIME 2.0, LEfSe and PICRUSt computational algorithms. Multiple machine learning (ML) models were trained to classify GM at T_0_ and T_1_. A statistically significant reduction of IL-6 (*p* < 0.001), TNF-α (*p* < 0.001) and IL-12RA (*p* < 0.02), citrulline (*p* value < 0.001) was reported at T_1_. GM global distribution and microbial biomarkers strictly reflected probiotic composition, with a general increase in Bifidobacteria at T_1_. Twelve unique KEGG orthologs were associated only to T_0_, including tetracycline resistance cassettes. ML classified the GM at T_1_ with 100% score at phylum level. Bifidobacteriaceae and *Bifidobacterium* spp. inversely correlated to reduction of citrulline and inflammatory cytokines. Probiotic supplementation during post-COVID-19 may trigger anti-inflammatory effects though Bifidobacteria and related-metabolism enhancement.

## 1. Introduction

Coronavirus disease 2019 (COVID-19), produced by the Severe Acute Respiratory Syndrome Coronavirus 2 (SARS-CoV-2), has been the biggest health emergency in the last hundred years, having a peerless impact over the whole world. Both in 2020 and 2021, COVID-19 has been the third cause of death in the United States, according to the US Centers for Disease Control and Prevention [1]. Typically, the disease is characterized by fever and respiratory symptoms and, particularly, by interstitial pneumonia [2]. However, many patients with COVID-19 frequently experience also digestive symptoms, such as diarrhea, vomit and abdominal pain. About the transmission route, the most relevant one is considered the direct exposure to infected patients, who can spread the virus through droplets or aerosols. However, SARS-CoV-2 is also detected in fecal samples. Generally, patients with respiratory infections often present associated intestinal dysfunction or intestinal secondary complications, with a more severe clinical course, suggesting a possible crosstalk between airways and gut [3,4]. This phenomenon has been also observed in COVID-19 patients [3]. New evidence suggests the possible presence of a specific lung microbiome signature, mainly characterized by predominant Bacteroidetes, Firmicutes, and Proteobacteria phyla compared to the most abundant Bacteroidetes and Firmicutes of the gut microbiome [5]. The gut-lung axis is supposed to be bidirectional, meaning that endotoxins and microbial metabolites deriving from the gut can impact the lung through blood and, on the other side, when inflammation firstly occurs in the lung, it can also affect the gut microbiota (GM) as well [6]. In fact, several studies have demonstrated that respiratory infections are associated with a change in the GM composition [7].

Regarding COVID-19, recent studies have found that GM richness was not restored to eubiotic conditions six months after primary SARS-CoV-2 infection, and patients with highest level of C-reactive Protein (CRP) and illness severity during the acute phase also showed lower post-convalescence richness, suggesting close correlations between inflammatory response and gut dysbiosis in COVID-19 [8]. Based on this evidence, GM modulation can be considered as a possible solution during and after recovering from COVID-19 acute phase. Interestingly, probiotics have been widely used as a mean to modulate GM, due to their safety and efficacy in specific settings [9]. During the first COVID-19 pandemic wave, there were no indications for the use of probiotics in COVID-19. However, the use of probiotic supplementation could have a sufficiently strong rational based on the previous history of use in humans with a beneficial effect on the immune system, due to the ability to boost the immune system and fight respiratory pathogens [10,11]. In fact, probiotics have been reported to act on our immune system in various ways, influencing both the innate and the adaptive immune system. Particularly, probiotics’ actions, based on non-immunological mechanism, may enhance the mucus layer of the gut barrier promoting its good functionality [12], but they can also act directly on our immune system, stimulating the production of cytokines such as TNF-α and IL-8, finally triggering a systemic innate immune response [13]. All these mechanisms could be beneficial in fighting viral infections and, particularly, COVID-19. The VSL#3^®^ is a multi-strain high potency probiotic mixture containing Lactobacilli, Bifidobacteria and *Streptococcus thermophilus*. Single species belonging to VSL#3^®^ mixture have been already reported to positively act on respiratory [10], gastrointestinal [14], influenza-like symptoms [15], or to enhance nasal innate immunity response to rhinovirus infection [16], or to boost humoral immune response following oral vaccination in healthy adults [17]. Theoretically, the combination of different strains in a mix, could amplify these effects, based on possible synergistic effects. In vitro, the exposure of Caco-2 cells to VSL#3-derived supernatant showed to positively impact on gut barrier function, increasing the function of tight junctions and accelerating the process of gut barrier repair [18]. Based on the possible gut barrier disruption after SARS-CoV-2 infection, we aimed with this pilot study to evaluate the effects of VSL#3^®^ on immune system in patients discharged after hospitalization for COVID-19, investigating the satisfaction with the supplementation and the safety and tolerability of the product, analysing the changes in inflammatory parameters, gastrointestinal symptoms, bowel movements, gut permeability and GM profiling.

## 2. Results

### 2.1. Study Population Features

Twenty patients were enrolled, but a patient was excluded from the final study phase because he prematurely discontinued the study due to consent withdrawal. Table 1 reports the baseline characteristics of the 19 patients: gender distribution, age, mean BMI, concomitant and previous other diseases, concomitant medications, previous surgery. Referring to the period of active COVID-19, the most frequent COVID-19-related symptoms at the onset of the infection were: fever (47.4%) and diarrhea (31.6%). For only three patients no symptoms were reported (Table 2).

During COVID-19, 47% of patients experienced gastrointestinal symptoms and 68% reported at least one concomitant disease. Hospitalization had a median duration of 14 days, with a range min-max between 2 and 78 days and a mean value of 18.05 ± 17.85 days. The median duration of infection positivity was 28 days, with a range min-max between 20.00 and 38.00 days and a mean value of 29.12 ± 15.78 days. The median time of days between negative swab and treatment initiation was 24 days, with a range min-max between 18 and 131 days and a mean value of 43.71 ± 38.23 days, while the days between COVID diagnosis and treatment initiation had a median of 56 days, with a range min-max between 33 and 187 days and a mean value of 72.05 ± 43.80 days.

#### Clinical Outcomes and Treatment Satisfaction Assessment

Regarding the Gastrointestinal Symptom Rating Scale (GSRS) total score, its calculation was effective only for 15 patients who answered all 15 questions both at baseline and after 8 weeks of supplementation. Regarding these 15 patients, 11 complained at least a gastrointestinal symptom at baseline. Among them, six showed gastrointestinal symptoms during COVID-19, while five not.

By comparing data registered at baseline and at the end of the supplementation, no statistically significant differences were found either in global and in specific GSRS item scores (Appendix A report mean scores and distribution of answers to all GSRS items before and after supplementation). However, looking at some single GSRS questionnaire items, some trends could be identified. Regarding the “passing gas or flatus” item, eight out of 13 patients reported symptoms at baseline and six out of eight improved at the follow-up visit. About the “constipation” and the “hard stools” items, all patients who declared discomfort at baseline experienced benefit after supplementation, while five of seven patients who declared “abdominal pain” at baseline benefited from the supplementation with the complete pain resolution at the follow-up.

No differences were found before and after supplementation in bowel habits. In fact, the mean number of evacuation and stool consistency, according to Bristol stool scale, did not show significant differences before and after supplementation, 1.24 ± 0.56 versus 1.15 ± 0.38 and 3.54 ± 0.924 versus 3.60 ± 0.834, respectively). Appendix A shows the frequency distribution of the individual items.

Regarding patients’ satisfaction on the purposed treatment, globally, 21.1% of patients were not satisfied by the supplementation, 36.8% were quite satisfied and 42.1% were very satisfied.

### 2.2. Immunological Response, Intestinal Barrier Integrity and Local and Systemic Inflammation

After 8-week probiotic supplementation, the levels of IL-6 (*p* < 0.001), TNF-α (*p* < 0.001) and IL-12RA (*p* < 0.02) were significantly reduced, with a variation of −5.75 ± 6.363 for IL-6, −4.85 ± 4.429 for TNF-α and −415.16 ± 708.954 for IL-12RA. No significant differences were reported for the other assessed cytokines (Figure 1).

Serum levels of citrulline were significantly reduced at the end of treatment (107.14 pg/mL), compared with baseline values (282.26 pg/mL) (*p* value < 0.001) (Appendix A), though within its physiological range (1750–6125 pg/mL) [19]. However, PV-1 (−0.22 ± 0.927, *p* value 0.322) and zonulin (−1.85 ± 21.746, *p* value 0.595) did not demonstrate significant variation after supplementation (Appendix A). Regarding faecal calprotectin, 16/19 patients showed normal values (≤50 μg/g) at the baseline. Similarly, 17/19 patients showed normal values of CRP (≤5 mg/dL) at baseline. However, no significant differences were found after supplementation for both markers of inflammation. In detail, a mean normal value of faecal calprotectin was measured for all patients at baseline (40.15 ± 81.91 μg/g) and after 8-weeks of supplementation (23.13 ± 30.45 μg/g). Similarly, CRP showed no significant variation after treatment, and mean values remained within the physiological range during all the observation (4.19 ± 10.38 mg/dL versus 2.22 ± 4.24 mg/dL, before and after treatment, respectively).

### 2.3. Gut Microbiota Profiling: Ecology, Predicted Functions and “Microbial Marker” Discovery

To assess the overall ecological differences of microbial communities pre and post probiotic supplementation, diversity algorithms were computed to compare sample sets at T_0_ and T_1_. Time points. The α-diversity was based on the following metrices: ChaoI to estimate the abundance of the individual samples belonging to the T_0_ and T_1_ classes; Shannon to assess both species richness and evenness, but with weight on the richness. Indeed, behind this latter metric there is the idea that more taxa you observe, more even their abundances are, the higher the entropy, or the higher the uncertainty of predicting which taxa you would see next; phylogenetic distance to evaluate phylogenetic diversity measures, that is the amount of the phylogenetic tree covered by the community; observed species to assess the observed ASVs counts up the number of ASVs you observe; Simpson to measure the degree of concentration when samples are classified into types; good’s coverage to estimate the percentage of total bacterial ASV represented in a sample (Figure 2) and dominance index to quantify the dominance of one or few species in a community. Greater values indicate higher dominance. Dominance indices are in general negatively correlated with alpha diversity indices (species richness, evenness, diversity, rarity).

A slight increase of α-diversity was observed at the T_1_ time-point compared to the T_0_, though this was not of statistical significance (*p* value > 0.05, ANOVA test), (Figure 2). Lower dominance index was reported for the gut microbiota communities at T_1_ compared to index computed at T_0_, corroborating the evidence of a more diverse microbiota at T_1_ (Appendix A).

The β-diversity, assessed by Bray-Curtis, Euclidian distance, unweighted and weighted UniFrac algorithms, did not provide statistically significant differences as ascertained by PERMANOVA test (*p* value > 0.05), between microbial communities at T_0_ and T_1_. time points (Appendix A). For each patient, fecal sample coupled comparisons, for global distribution description of gut microbiota at T_0_ and T_1_, respectively, were reported at phylum (L2), family (L5) and genus (L6) levels but not differential profiles were detected (Appendix A).

However, for the overall coupled global sample sets’ comparison, at T_1_ versus T_0_, the ASVs relative abundance differences at taxonomic phylum level relied on Firmicutes, Euryarchaeota, Bacteroidetes, Proteobacteria, Actinobacteria, with an increase in Actinobacteria and Bacteroidetes and a decrease in Proteobacteria, respectively, at the T_1_ time-point, though these were not of statistical significance (Appendix A). At the taxonomic family level, Enterobacteriaceae, Clostridiaceae, Peptostreptococcaceae appeared higher in the gut microbiota of COVID-19 patients at T_0_, while Streptococcaceae, Ruminococcaceae and Bifidobacteriaceae were more abundant in the gut microbiota at T_1_, the latter being statistically significant (*p* < 0.05) (Figure 3).

At the taxonomic genus level (L6), *Ruminococcus*, *Oscillospira*, *Streptococcus* and *Bifidobacterium* were more abundant at T_1_ than at T_0_, with *Bifidobacterium* being statistically significant (*p* < 0.05) (Figure 4).

To infer taxonomic differences between the GM of the patients at T_0_ and T_1_, before and after probiotic supplementation, in terms of potential biomarkers, the LefSe algorithm was exploited and six top-ranking ASVs overall characterizing the GM at the T_1_ time-point were identified (Figure 5). In particular, the analysis provided T_1_-related specific microbial biomarkers (*p* ≤ 0.05), such as Actinobacteria (L2, phylum), Bifidobacteriales (L4, Order), Bifidobacteriaceae and Enterococcaceae (L5, family); *Bifidobacterium* and *Enterococcus* (L6) (Figure 5).

To investigate the GM of post-COVID-19 patients by potential microbial marker searching, its probiotic-linked profiling at T_1_ was investigated by a computational analysis based on multiple ML models trained to classify the patients’ GM at T_1_ versus T_0_ for each taxonomy level of the corresponding ASVs, hence identifying “important” features. The ML model based on K Neighbors Classifier had an accuracy of 87.5% in distinguish between T_1_ and T_0_ patients’ gut microbiota at Phylum level (Figure 6A), while the model based on SGD Classifier had an accuracy of 58.33% in distiguish the two gut microbiota at family and genus levels (Figure 6B,C). However, the accuracy value rised up to 100.0% by K Neighbors Classifier when referring just to T_1_ patients’ gut microbiota classification at phylum level, and to the 66.7% at both family and genus levels. In all cases, the important features of the T_1_ patient group were related to *Bifidobacterium*-related taxa, actually one of the main components of the administered probiotic, and to short chain fatty acids (SCFA) producer bacteria.

The PICRUSt algorithm, exploited to identify predicted functional signatures, allowed us to identify 97 KEGG orthologs (KOs), filtered by statistically significance, up or down represented for the T_0_- and T_1_-related gut microbiota ASV datasets. Amongst this KOs set, 12 KOs were mainly associated to bacterial virulence and antibiotic resistance and resulted unique, that is exclusivetely associated to the time point T_0_ and absent at T_1_. Indeed, the KOs were: K16958 (i.e., L-cystine transport system permease protein), K14988 (i.e., two-component system, NarL family, secretion system sensor histidine kinase SalK), K00720 (i.e., ceramide glucosyltransferase), K01274 (i.e., β-Ala-Xaa dipeptidase), K01819 (i.e., galactose-6-phosphate isomerase), K02531 (i.e., Transcription factor), K02771 (i.e., fructose PTS system EIID component), K12294 (i.e., LytTR family, sensor histidine kinase ComD), K12295 (i.e., two-component system, LytTR family, response regulator ComE), K18216 (i.e., ATP-binding cassette, subfamily B, tetracycline resistant protein), K18217 (i.e., ATP-binding cassette, subfamily B, tetracycline resistant protein), K18830 (i.e., HTH-type transcriptional regulator/antitoxin PezA) (Appendix A).

Moreover, in correlation heatmaps, Spearman’s algorithm was used to examine the association between features (e.g., biochemical markers and ASVs) and only statistically significant correlations (FDR adjusted *p* values < 0.05) were reported. Spearman’s correlation highlighted significant negative correlation between citrulline and ASV at L2 (phylum Actinobacteria), L5 (family Bifidobacteriaceae) and L6 (genus *Bifidobacterium*) (FDR adjusted *p* values < 0.05). Also, significant negative correlation between IL-6 and ASV was observed but only at L2 (phylum Bacteroidetes) and L5 (family Methanobacteriaceae) (*p* values < 0.05) (Figure 7). No statistically significant correlations were identified between KOs and biochemical markers.

## 3. Discussion

COVID-19 pandemic dramatically changed our lives. The impact of this disease goes beyond the effects that we see during the active phase, as long-COVID syndrome has been described, that could deeply impair health many months after COVID-19.

In this small pilot study, designed before that long-COVID [20,21] was described, we aimed to explore if the modulation of GM through probiotic supplementation could be able to contrast COVID-19-induced alterations in term of barrier integrity and immune response. Considering the pivotal role of gut barrier in maintaining local and systemic immune homeostasis, we explored GM ecological and functional inferred profiles, indirect markers of gut barrier integrity and systemic cytokine profile, together with clinical monitoring of gastrointestinal symptoms pre- and post-probiotic supplementation.

Since the first outbreak in 2019, SARS-CoV-2 has undergone multiple variants over time. These variants have developed mutations capable of conferring higher transmissibility or antigenicity. However, the effect of probiotic supplementation demonstrated in this paper is not expected to change with different SARS-CoV-2 variants, as it is mostly related to probiotics contained in the mix rather than the interaction with the virus.

Interestingly, most patients at baseline (T_0_), which was a post-COVID free-of- supplementation timepoint, complained at least one gastrointestinal symptom, independently from the presence of gastrointestinal symptoms during COVID-19 active infection. Furthermore, considering that in our cohort patients with a previous known chronic gastrointestinal disease were excluded, this data could suggest a role of COVID-19 in triggering post-infectious functional gastrointestinal disorders [22,23,24,25].

Notwithstanding a high prevalence of gastrointestinal symptoms at baseline, probiotic supplementation did not significantly improve symptoms at the end of treatment, probably because of the small sample size and the heterogeneity of reported gastrointestinal symptoms. However, looking at the single items of the GSRS questionnaire, some trends toward improvement could be highlighted even if not statistically significant, particularly regarding flatulence, constipation, and abdominal pain.

The presence of these gastrointestinal symptoms at baseline was not associated with active gastrointestinal inflammation as demonstrated by normal value of CRP and faecal calprotectin at baseline. Similarly, no gut barrier dysfunction was detected at baseline through zonulin evaluation, suggesting that after the acute phase of COVID-19 infection, the epithelium in the gut rapidly repaired, consistently with previous reports [26,27,28].

However, the systemic activation of the immune response seemed to continue after clinical recovery, as the serum level of cytokines was high at baseline. In this scenario, the use of probiotic supplementation could have an impact on systemic cytokine profile, as at T_1_, after 8-weeks of probiotic supplementation, the serum levels of IL-12RA, IL-6, TNF-α were significantly reduced.

However, this result could also reflect the natural history of COVID-19, as at the T_1_ COVID-19 acute form is more remote compared to baseline, thus we should take in account also the hypothesis of a natural cytokine decrease due to the overcome of the acute disease, more than an effect related to the probiotic supplementation. Clearly, the absence of a placebo group in our study represents a major limitation, as it does not allow us to definitely interpret this data as natural history of disease or a real immunological impact of the probiotic supplementation.

Some studies about SARS-CoV-2 inflammation have already shown how, during infection, serum levels of IL-1b, TNF-α and other cytokines were higher in infected patients than in healthy adults [29] and the highest IL-6 and TNF-α levels correlated with disease severity [30,31], suggesting cytokine levels as good indicator of therapeutic goal [32]. This evidence may support the hypothesis that probiotic supplementation could have a positive impact on cytokine profile reduction.

Before and after probiotic supplementation, ecology of the microbial ecosystems appeared characterized by an increase in richness, though this was not of statistical significance, but were represented by an increased quantity of Streptococcaceae, Ruminococcaceae and Bifidobacteriaceae at T_1_, the latter being statistically significant and corroborated at L6 by the increase of the *Bifidobacterium* spp. Remarkably, the LEfSe algorithm, exploited for searching of potential microbial biomarkers, identified six top-ranking ASVs overall characterizing the GM at the T_1_ time-point. Amongst the others, Actinobacteria (L2, phylum), Bifidobacteriales (L4, Order), Bifidobacteriaceae (L5, family); Bifidobacterium were consistent with the probiotic composition. Additionally, based on the ML model classification analysis, the two ecological microbial ecosystems at T_0_ and T_1_ were correctly classified at L2 level by 87.50% score and the microbial ecosystem at T1 was predicted with a 100.00% score, based on the feature Actinobacteria, phylum of the Bifidobacterium spp. At level L5 and L6, behind specific *Bifidobacterium* family- and genera-related, the most important features resulted overall SCFAs producers, well characterizing the GM of the patients at T_1_, compared to T_0_, regardless a lower performance score (i.e., 58.33) compared to the values based on L2 feature. Based on these results, *Bifidobacterium* spp. seem to be the most functionally active component of the multi-strain probiotic product as it strongly characterized the GM ecology after supplementation. Indeed, further metabolomics-based investigations could unveil biochemical pathways of probiotic metabolite-derived, as already performed for functional assessment of probiotic supplementations in other diseases [33].

Amongst the 97 GM functional KOs signatures, up or down represented for the T_0_- and T_1_-related ASV datasets, 12 were unique Kos associated to the only pre-treatment time point T_0_, but completely absent at T_1_. Particularly, the K18216 and the K18217, both defined as ATP-binding cassette, subfamily B, tetracycline resistant protein, were associated to the metabolic pathways TetAB(46), a predicted heterodimeric ABC transporter conferring tetracycline resistance and to steB/tetB46, respectively, apparently conferring ABC transporter tetracycline resistance in a member of the oral microbiota in the first case [34] and acting as ATP-binding cassette, subfamily B, tetracycline resistant protein in the other [24]. This data could suggest a possible role of probiotic supplementation in enhancing the clearance of antibiotic-resistant genes in GM of patients that received antibiotic therapy during COVID-19. In fact, 10/19 patients had been treated with single or combined (subsequent or concomitant) antibiotics -including amoxicillin/clavulanate, ceftriaxone, piperacillin/tazobactam, clarithromycin or azithromycin- for prophylaxis or treatment of bacterial complication of COVID-19 during the hospitalization.

Interestingly, Spearman’s correlation highlighted significant negative correlation between citrulline and ASV at L2 (Actinobacteria), L5 (Bifidobacteriaceae) and L6 (*Bifidobacterium*) (FDR adjusted *p* values < 0.05). Also, significant negative correlation between IL-6 and ASV was observed but only at L2 (Bacteroidetes) and L5 (Methanobacteriaceae) (*p* values < 0.05).

In this context, we better understand the apparently surprisingly reduction of serum citrulline after supplementation. In fact, in the literature, citrulline has been used as a quantitative enterocyte mass marker [27] and it generally decreased during an acute mucosal damage and subsequently increased because of epithelial healing. However, the decrease of citrulline in our cohort after supplementation was not probably related to gut barrier damage, as it was not related to significant changes in tight junctions’ function as demonstrated by no change in serum zonulin. Furthermore, notwithstanding a significant reduction compared to baseline, both baseline and T_1_ values remained in physiological range. This apparently unexpected curve of serum citrulline did not seem to be related to variation on enterocyte mass, but in this functional context, its modification could be related to a shift in GM ecology and bacterial metabolism. In fact, we showed an inverse correlation between serum citrulline and *Bifidobacterium* spp. probably due to specific amino acids-linked metabolic pathways, as already reported for GM *Akkermansia* [25] and not to physiological effects.

## 4. Materials and Methods

### 4.1. Study Design and Aims

This open-label, single-center, single-arm, pilot, interventional study was conducted in compliance with the independent Ethics Committee/Institutional Review Board (EC/IRB)’s recommendation, informed consent regulations, Declaration of Helsinki, ICH GCP Guidelines and this Study Protocol. The protocol Probiotics against COronavirus: a pRoof-of-concept, Open-LabeL, single-Arm, single-centre clinical study in patients with COVID-19- COROLLA study was approved by the Ethical Committee of Fondazione Policlinico Universitario A. Gemelli IRCCS (protocol version 2.0, 28 September 2020). All participants provided written informed consent for all clinical and experimental procedures and publication of the results, before any study step. All subjects participated to three study visits: week -2 (screening), week 0 (start of treatment), week 8 (end of treatment). Patients were asked to collect blood and fecal samples and to underwent clinical and gastrointestinal symptoms evaluation before and after 8 weeks of treatment. Eligible patients were asked to continue with their dietary habits during the whole course of the trial.

The primary aim of the study was to evaluate the effect of VSL#3^®^ on the immunological response through a serum cytokine profile, including IL-6, IL-1, IL-2R, IL-8, IL-10, TNF-α, before and after VSL#3^®^ supplementation. Secondary aims relied on the evaluation of the VSL#3^®^ effects on gastrointestinal symptoms, bowel movements, gut permeability and gut microbiota profile. An overall questionnaire on treatment satisfaction, safety and tolerability of the product was also administered.

All enrolled patients were assigned to the following probiotic supplementation: VSL#3^®^ 450 billion sachets, 1 sachet twice a day for 8 weeks. VSL#3^®^ (lot number 909031) was provided by Actial Farmaceutica. The probiotic mix contained four strains of lactobacilli spp.: *Lactobacillus paracasei* BP07, *Lactobacillus plantarum* BP06, *Lactobacillus acidophilus* BA05 and *Lactobacillus helveticus* BD08 (previously identified as *L. delbrueckii* subsp. *bulgaricus*); three strains of bifidobacteria: *Bifidobacterium animalis* subsp. *lactis* BL03 (previously identified as *B. longum*), *Bifidobacterium animalis* subsp. *lactis* BI04 (previously identified as *B. infantis*) and *Bifidobacterium breve* BB02; a strain of *Streptococcus thermophilus* BT01.

The study population consisted of 20 patients discharged after hospitalization for COVID-19 and in follow-up at the post-COVID-19 Day Hospital of the Fondazione Policlinico Universitario A. Gemelli IRCCS for SARS-CoV-2 infection during the second pandemic wave in Italy, with a previous positive nasopharyngeal swab, performed at the dedicated “COVID-19 hotel” nursery or at home. Patients were considered eligible for the study if they had an age between 18 and 80 years and if they were previously discharged after hospitalization for COVID-19 and if they had a second negative nasopharyngeal swab, proving recovery from COVID-19 acute phase. Exclusion criteria were: patients discharged after hospitalization addressed to a Nursing and Residential Care Facility; ascertained intestinal organic diseases, including inflammatory bowel diseases (Crohn’s disease, ulcerative colitis, infectious colitis, ischemic colitis, microscopic colitis); presence of any severe organic, systemic or metabolic disease (particularly significant history of cardiac, renal, neurological, psychiatric, oncological, endocrinological, metabolic or hepatic disease), or abnormal laboratory values that would be deemed clinically significant by the Investigator; active malignancy of any type, or history of a malignancy, either surgically removed and with evidence of recurrence for at least five years before the study enrolment; use of probiotics or prebiotics during the last two weeks before screening; inability to be conformed to the protocol; pregnancy or breastfeeding; participation in other investigational studies or treatment with any investigational drug within the previous 30 days.

### 4.2. Measures of Clinical Outcomes

The presence of gastrointestinal symptoms before and at the end of the study was investigated through the GSRS questionnaire [35] and by recording the number of bowel movements and stool consistency by the Bristol stool form scale [36] in the three days before each visit. The GSRS questionnaire includes 15 questions each for one different gastrointestinal symptom, referring to five symptom clusters: reflux, abdominal pain, indigestion, diarrhea and constipation. For each symptom, the patient is asked to choose the corresponding grade of intensity on a 7-point Likert scale (from no symptoms to unbearable symptoms).

A visual analogue scale (VAS) was used to register patients’ satisfaction about the supplementation at the end of treatment. We also collected information about the previous COVID-19, including the most prevalent symptoms, duration of hospitalization and specific therapies administered.

### 4.3. Measures of Laboratory Outcomes

The analysis included serum cytokines (interleukins; IL-1β, IL-12RA, IL-6, IL-8 and tumor necrosis factor, TNF-α) measured using Ella™ (Bio-Techne, Oxford, UK) [37,38], and serum markers of intestinal barrier integrity and permeability including zonulin, citrulline and plasmalemmal Vesicle Associated Protein-1, PV-1 assessed by ELISA test [26,27,28,39]. Blood samples were collected before (T_0_) and after the 8 weeks (T_1_) of treatment in a vial without anticoagulant to obtain serum.

Faecal calprotectin was used as a marker of intestinal inflammation [40] and C-reactive protein (CRP) as a marker of systemic inflammation. Stool samples for faecal calprotectin determination were collected preferentially on the same day of visit. In case this was not possible, also samples collected the day before were accepted. Samples were stored in home fridge by the patient before they were brought to the center. Both faecal calprotectin and CRP were collected at baseline and at the end of the 8-week treatment. Analysis was performed locally according to Biochemistry Laboratory procedures.

### 4.4. Statistical Analysis of Metadata

No data were available in the literature on the use of probiotics for immune modulation during COVID-19 at the time of the protocol design. Thus, the sample size was not based on a formal statistical sample size calculation, but was considered appropriate for the study purposes, also considering the estimated enrolment rate at the site. We estimated that 20 patients would be enough to individuate a trend that could allow formal sample size calculation for eventual further studies. Regarding statistical methods, a *p* value <0.05 was considered statistically significant. Quantitative variables were represented as statistical mean and standard deviation (sd) or median and first and third quartiles (Q1–Q3). Categorical variables were represented as absolute frequency and percentage (%). The paired T_0_–T_1_ Student *t* test and corresponding Wilcoxon non-parametric test were used to evaluate the variation between baseline and end-of-study visit.

### 4.5. Faecal Microbiota Analyses

Forty stool samples were planned to be collected at T_0_ (week 0) and T_1_ (week 8) for gut microbiota profiling of each 20 patients, through 16S rRNA sequencing. Specimens were collected preferentially on the same day of visit or the day before, if necessary, and stored in the freezer until visit. All samples were then stored at the −80 °C until processing. After selection for only coupled T_0_–T_1_ stool samples, 36 samples were processed, because of two lacking samples at T_0_ and T_1_, respectively.

#### 4.5.1. Bacterial DNA Extraction from Stools and 16S rRNA Targeted-Metagenomics

Stools were processed in in a strictly controlled, separate and sterile workplace. Briefly, 200µg of each sample were resuspended in CTAB buffer. This suspension was used to extract DNA by using Danagene Microbiome Stool DNA kit (DanaGen-Bioted, S.L., Barcelona, Spain) according to manufacturer’s instruction [41]. Quality and concentration of the extracted DNA were evaluated before amplifying the variable regions V3–V4 from the bacterial 16S rRNA gene (∼460 bp) by using the following primers: V3_Next_For: 5′-TCGTCGGCAGCGTCAGATGTGTATAAGAGACAGCCTACGGGNGGCWGCAG-3′ and V4_Next_Rev: 5′-TCTCGTGGGCTCGGAGATGTGTATAAGAGACAGGACTACHVGGGTATCTAATCC-3′ [42]. Amplicons were purified by using Agencourt AMPure XP beads (Beckman Coulter, Brea, CA, USA) and then barcoded with Nextera XT index (Illumina) according to Illumina rRNA Amplicon Sequencing protocol (Illumina). Each indexed amplicon was equimolarly diluted and the final pool was properly prepared for the paired ends sequencing (2 × 300 bp, v3 chemistry, Illumina) on the Illumina MiSeq instrument (Illumina, San Diego, CA, USA). To increase degree of base diversity, the internal control PhiX v3 (Illumina) was added to the library [42].

#### 4.5.2. Biocomputational and Statistical Analysis for GM Profile Analysis and Patients’ Metadata Correlation

Paired-end sequencing reads in fastq format were analyzed using QIIME2 [43]. Samples characterized by reads number < 20,000 were excluded, hence finally providing 34 samples totally, with 17 sample coupled for each T_0_-T_1_ paired time-point. The QIIME2 plugin for DADA2 [44] was used for quality control, denoising, chimera removal, trimming and construction of the Amplicon Sequence Variant (ASV) table. The taxonomy was assigned by using a Naive Bayes model pre-trained on Greengenes 13_8 [45,46] through the QIIME2 plugin q2-feature classifiers [47]. Unassigned reads were filtered out while the ASV table was normalized using the Cum Sum Scaling (CSS) methodology [48]. Alpha-diversity was computed by skbio.diversity using analysis of variance (ANOVA test). A comparison of ASV taxonomic differences at phylum (L2), family (L5), and genus (L6) levels for each T_0_–T_1_ couple was provided. Kruskal-Wallis test was applied to compare taxonomic differences at L2, L5 and L6 for the entire T_0_ and T_1_ datasets, respectively. All ecological statistical analyses were performed using Python 3.7. Three different levels of statistical significance were identified based on different *p* values (*p* ≤ 0.001) and false discovery rate (FDR) thresholds (*p* ≤ 0.05, *p* ≤ 0.001) and two of them were corrected for multiple hypothesis testing by FDR method [49]. Both statistical tests (ANOVA and Kruskal–Wallis) were chosen to provide information on whether the parametric and non-parametric tests would lead to similar or different conclusions. The results of both tests were consistent, providing a more nuanced understanding of the data, approximately normally distributed (Shapiro-Wilks test), and with homogeneity of variances (Levene’s test) and a continous dependent variable. The algorithm for high-dimensional biomarker discovery and explanation, based on linear discriminant analysis (LDA) effect size (LEfSe) [50], was employed to identify ASV features that resulted statistically different among T_0_ and T_1_ GM groups. Specifically, a non-parametric factorial Kruskal–Wallis (KW) sum-rank test followed by Wilcoxon rank-sum test were used. Finally, LDA estimated the effect size of each differentially abundant ASV.

Phylogenetic Investigation of Communities by Reconstruction of Unobserved States (PICRUSt), exploiting the Kyoto Encyclopedia of Genes and Genomes (KEGG) orthologs (KO) database were used to determine ASVs and their microbiome’s functional potential and the unique KO associated to pre- and post-treatment.

In correlation heatmaps, Spearman’s correlation was used to examine the association between features (e.g., biochemical markers and ASVs) and only statistically significant correlations (FDR adjusted *p* values < 0.05) were reported.

Positive and negative association between ASVs and markers of immunological response and intestinal barrier integrity which resulted significantly modified after treatment were selected according to mean Spearman correlation index ρ (rho) >0 or <0 in positively and negatively, respectively associated markers.

#### 4.5.3. Machine Learning Models

Multiple machine learning models were trained for the classification task T_1_ versus T_0_ at each taxonomy level and for the corresponding ASVs. Multiple machine learning (ML) models were trained for the classification tasks. The pipeline consisted of a 10-fold cross-validation with a train-test split of 70–30%. To evaluate the model, the global and the single-class accuracies were considered. The models tested were Logistic Regression, SGD Classifier, Logistic Regression CV, Hist Gradient Boosting Classifier, Random Forest Classifier, Extra Trees Classifier, Gradient Boosting Classifier, Bagging Classifier, Ada Boost Classifier, XGB Classifier, XGBRF Classifier, MLP Classifier, Linear SVC, SVC, Gaussian NB, Decision Tree Classifier, Quadratic Discriminant Analysis, K Neighbors Classifier, and Gaussian Process Classifier. An explain ability algorithm based on a permutation performance with 1000 repetitions was followed.

## 5. Conclusions

Probiotic supplementation in the immediate post-COVID-19 could have a positive impact on immunological profile, reducing pro-inflammatory systemic cytokines through a shift in GM ecology.

GM profiling, either at ecological and inferred functional level, was extremely correlated to pre-and post- probiotic supplementation. A predicted model, based on ML classification was indeed predictive of the T_1_ with 100% score, based on the same ASVs (i.e., Actinobacteria) characterizing the microbial ecosystem in term of ecology, microbial biomarker LefSE prediction and strictly consistent with probiotic components (i.e., *Bifidobacterium* spp.), proposing the GM as the master regulator of the immunological host response to probiotic supplementation. *Bifidobacterium* spp. are metabolically active, and they may correlate with a reduction on citrulline levels. These preliminary results, which should be confirmed in placebo-controlled clinical studies and further sampling scale up, provide us with a first insight in the possible mechanism of action of a multi-strain probiotic in humans.

## Figures and Tables

**Figure 1 ijms-24-06623-f001:**
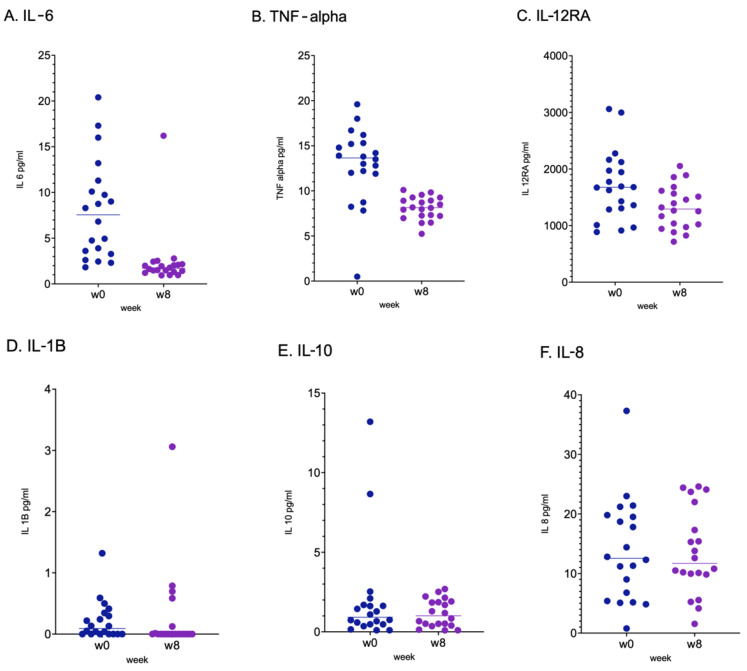
Values of cytokines detected at baseline (Week 0, W0) and at the end of the study (Week 8, W8). *p* values for IL-6 (**A**), TNF-α (**B**) and IL-12RA (**C**) were *p* < 0.001 and *p* < 0.02, respectively. *p* values for IL-1B (**D**), IL-10 (**E**) and IL-8 (**F**) were not significant. Paired data were compared by Wilcoxon test after verification of non normal distribution by graphical qqplot representation. The comparisons are reported at T_0_ (Week 0, w0) and T_1_ (Week 8, w8), respectively.

**Figure 2 ijms-24-06623-f002:**
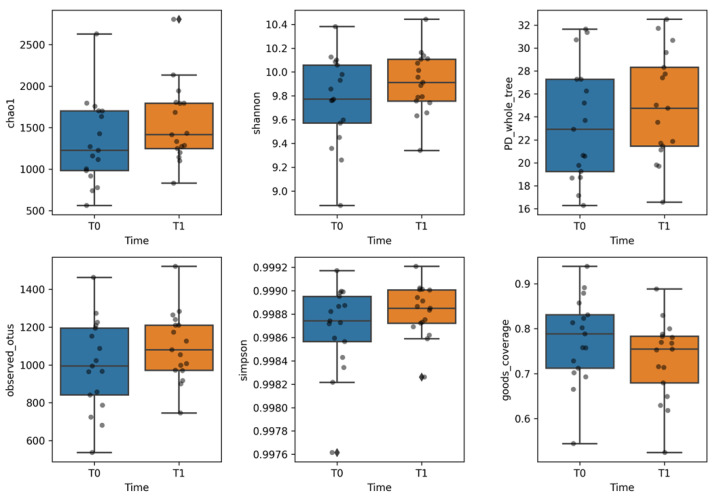
Evaluation of the gut microbiota ecology assessed by α-diversity of faecal sample sets at T_0_ and T_1_ time points, corresponding to pre- and post-probiotic supplementation, based on Chao-1, Shannon, observed species, phylogenetic distance, goods coverage and Simpson diversity metrices. No statistical significance was observed (*p* value > 0.05), based on both ANOVA and Kruskal-Wallis tests. Both statistical tests were confirmed by Shapiro-Wilk and Levene’s tests, respectively. The comparison are reported at T_0_ (Week 0) and T_1_ (Week 8), respectively.

**Figure 3 ijms-24-06623-f003:**
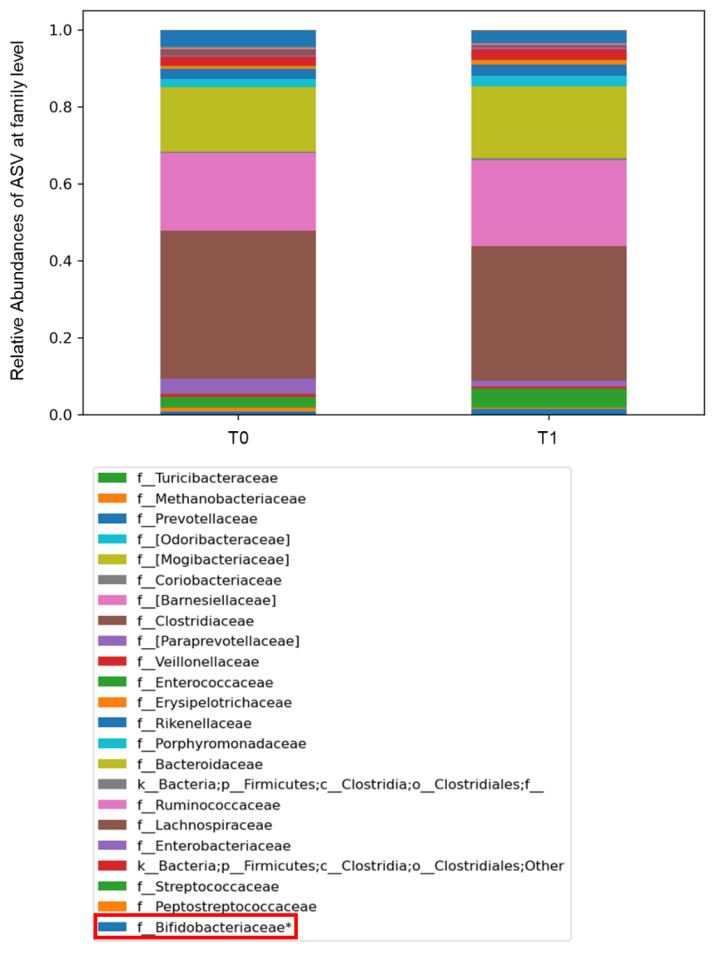
Histograms representing the relative abundancies of Amplicon Sequence Variant (ASV) distributions in the gut microbiota of patients at T_0_ and T_1_, represented at taxonomic family level (L5). The bacterial families were compared by Kruskal–Wallis test. The box in red refers to the family Bifidobacteriaceae, relevant for the composition of the administered probiotic, including *Bifidobacterium*, and filtered by statistical significance (* *p* value < 0.001).

**Figure 4 ijms-24-06623-f004:**
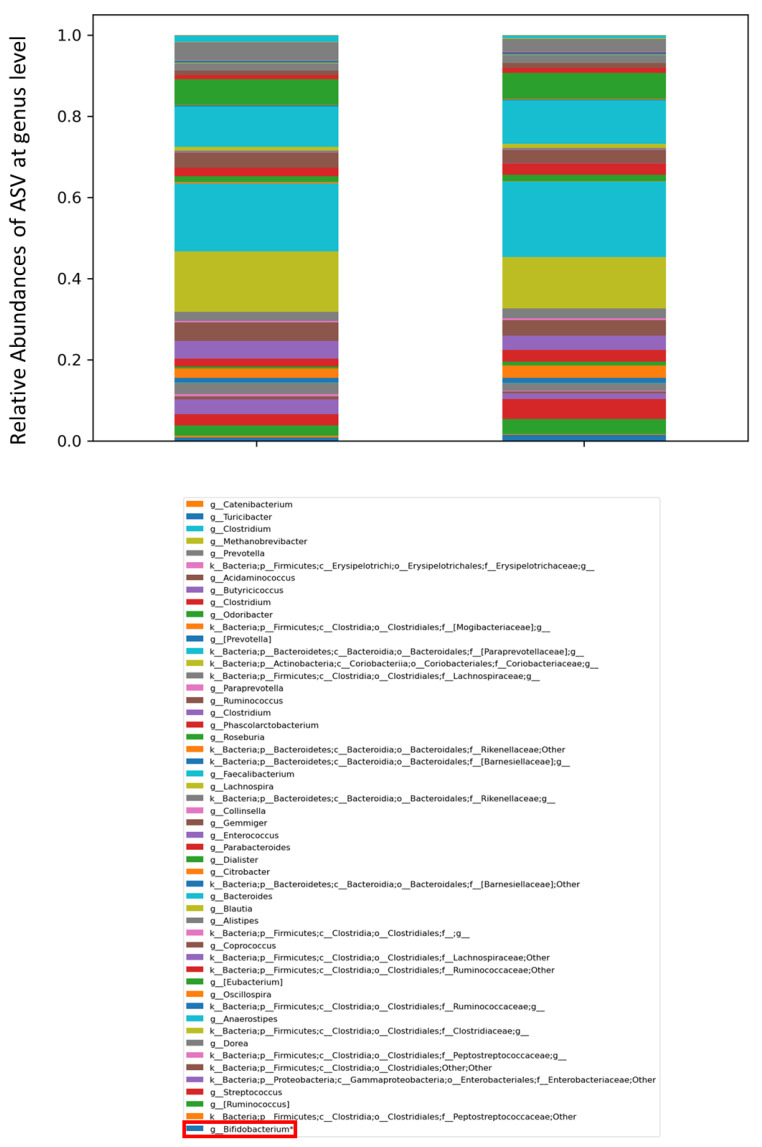
Histograms representing the relative abundancies of Amplicon Sequence Variant (ASV) distributions in the gut microbiota of patients at T_0_ and T_1_, represented at taxonomic genus level (L6). The bacterial genera were compared by Kruskal-Wallis test. The box in red refers to the genus Bifidobacterium, relevant for the composition of the administered probiotic and filtered by statistical significance (* *p* value < 0.001).

**Figure 5 ijms-24-06623-f005:**
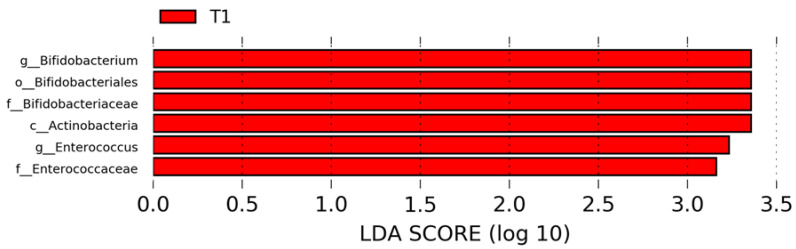
LEfSe analysis for high-dimensional biomarker discovery and explanation, based on linear discriminant analysis (LDA) effect size (LEfSe), employed to identify ASV features that resulted in statistical differences among T_0_ and T_1_ gut microbiota groups. Histogram represents the LDA scores until the taxonomy level genus (L6) filtered by statistical significance between the two groups. In red are represented microbial biomarkers for the patients at T_1_.

**Figure 6 ijms-24-06623-f006:**
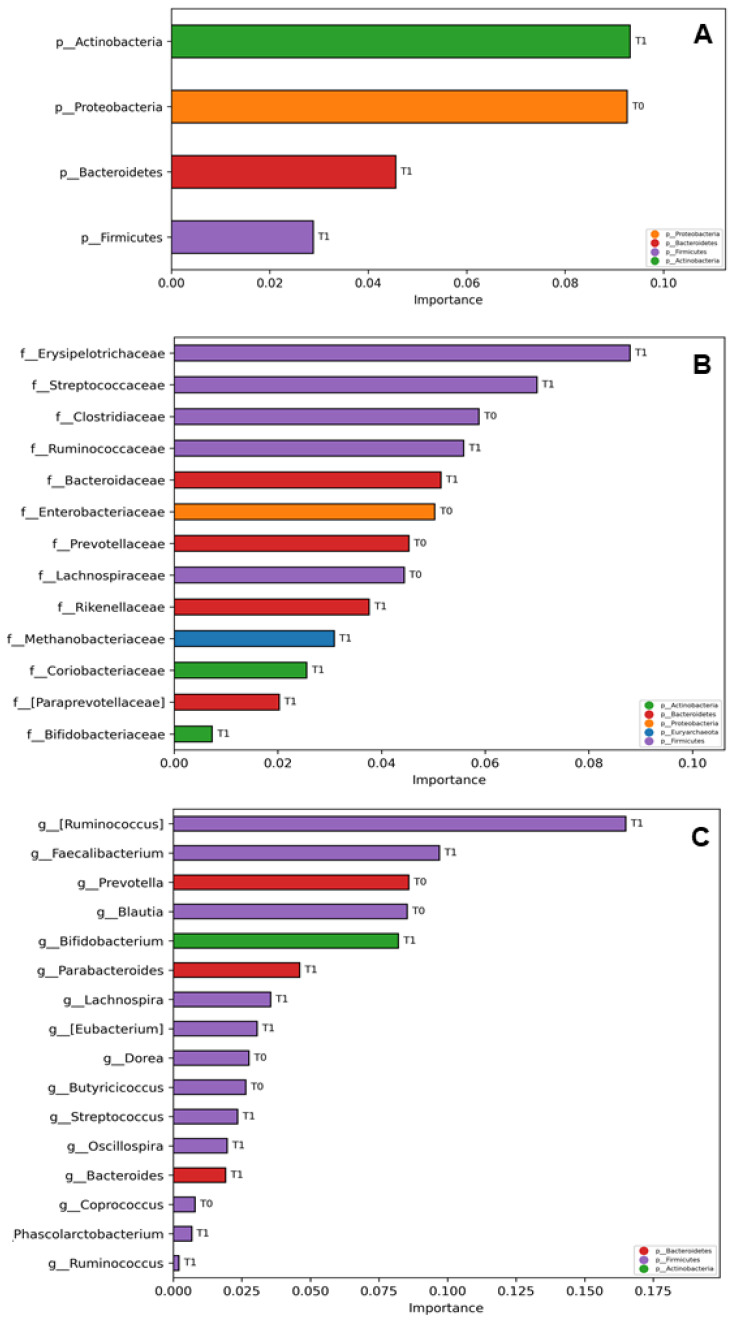
Multiple machine learning (ML) models trained to classify the patients’ gut microbiota at T_1_ versus T_0_ for each taxonomy level of the corresponding ASVs, selected by the model classification analysis as “important” features. The bars represent the importance scores of each ASV in the prediction of models. Important ASVs for model prediction are represented at Phylum level (L2) (**A**); Family level (L5) Pael (**B**); and at Genus level (L6) (**C**). The labels T_0_ and T_1_ represent the group of patients for which the relative ASV resulted more representative. The models able to classify T_1_ versus T_0_ gut microbiota at phylum level (L2) was K Neighbors Classifier and at family level (L5) and genus level (L6) was SGD Classifier with 87.50% and 58.33% of accuracy for L2 and L5, L6, respectively).

**Figure 7 ijms-24-06623-f007:**
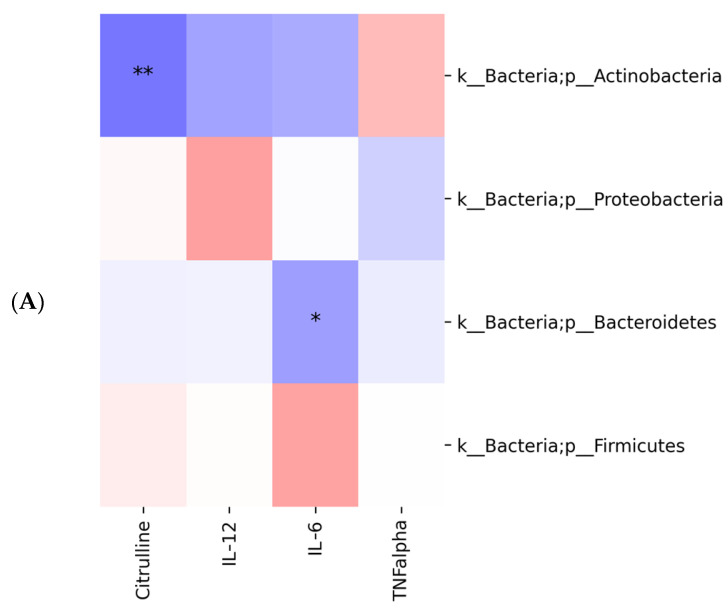
Spearman’s correlation between citrulline, IL-12, IL-6, TNF-a and ASV. Spearman’s correlation is represented at L2-phylum (**A**), L5-family (**B**) and L6-genus (**C**) levels (FDR adjusted *p* values < 0.05). The statistical significance is represented with * for *p*-value < 0.05, and ** for *p*-value < 0.01.

**Table 1 ijms-24-06623-t001:** Patient characteristics at baseline.

		Summary Statistics	Tot (*n* = 19)
Sex	Female	%, *n*	31.6% (6/19)
	Male	%, *n*	68.4% (13/19)
Age		*n*	19 (100.0%)
		Mean ± SD	55.00 ± 8.56
		Median (Q1-Q3)	54.00 (52.00–61.00)
		Min-Max	33.0–70.0
		Missing	0
BMI		*n*	19 (100%)
		Mean ± SD	26.53 ± 4.20
		Median (Q1-Q3)	25.56 (23.90–28.70)
		Min-Max	18.5–40.9
		Missing	0
Other diseases (concomitant and previous)		%, *n*	68.4% (13/19)
Concomitant diseases	Anxious depressive syndrome	%, *n*	4.8% (1/21)
	Benign prostatic hypertrophy	%, *n*	14.3% (3/21)
	Diabetes	%, *n*	9.5% (2/21)
	Gastroesophageal reflux	%, *n*	14.3% (3/21)
	Hiatal hernia	%, *n*	4.8% (1/21)
	Hypercholesterolemia	%, *n*	4.8% (1/21)
	Hypertension	%, *n*	28.6% (6/21)
	Insomnia	%, *n*	4.8% (1/21)
	Minor beta-thalassemia	%, *n*	4.8% (1/21)
	Osteoporosis	%, *n*	4.8% (1/21)
	Tachycardia	%, *n*	4.8% (1/21)
Previous diseases	Cerebral Ischemia	%, *n*	20.0% (1/5)
	Chlamydial pneumonia	%, *n*	20.0% (1/5)
	Previous HBV	%, *n*	20.0% (1/5)
	Thyroiditis	%, *n*	20.0% (1/5)
	Ulna and radius fractures	%, *n*	20.0% (1/5)
Concomitant medication		%, *n*	78.9% (15/19)
Previous surgery		%, *n*	42.1% (8/19)

**Table 2 ijms-24-06623-t002:** COVID-19 related symptoms during hospitalization.

	Summary Statistics	Tot (*n* = 19)
Fever	%, *n*	47.4% (9/19)
Cough	%, *n*	21.1% (4/19)
Anosmia	%, *n*	10.5% (2/19)
Diarrhea	%, *n*	31.6% (6/19)
Other symptoms	%, *n*	36.8% (7/19)

## Data Availability

All raw sequences have been archived in NCBI database: project code PRJNA925216 (https://www.ncbi.nlm.nih.gov/bioproject, accessed on 18 January 2023).

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
