# Peer review of "Ecology and Machine Learning-Based Classification Models of Gut Microbiota and Inflammatory Markers May Evaluate the Effects of Probiotic Supplementation in Patients Recently Recovered from COVID-19"

_ijms, 2023, doi:10.3390/ijms24076623_

Round 1
Reviewer 1 Report
Laterza and colleagues analyze the effect of gut microbiota on patients’ recovery from COVID-19 after using a proprietary probiotics formulation. Conflicts of interest are clearly stated. I found the work interesting, but it needs substantial improvements.
Comments:
I have general considerations on figures (both main figures and supplementary figures). Some are also highlighted explicitly in other points:
– Can the authors overlay single points representing single measurements for all graphs with only summary statistics (boxplot, bar charts, es. Fig 1, 2, S1…)?
– In many figures is not reported the unit of measurement on the y-axis.
– Figure legends must be exhaustive, indicate what the figure represents, and report information that helps the reader understand.
– Many figures look like they are copied and pasted straight from the analysis pipelines. A bit of editing would improve the overall perceived quality of the work.
Other issues:
– Page 3, lines 128 - 134. The authors state that some “trends” are emerging. Although it might be true, giving any reasonable scientific evaluation is problematic, considering the small number of enrolled patients and the lack of statistical support. This is also the main issue with this work, as the authors correctly point out.
– Figure 1. What is the y-axis representing? The unit of measurement needs to be included. Can the authors improve the readability by showing a point for each measurement (in addition to single box plots) and indicating directly on the graph the significant comparisons? Please also add what statistics you used for calculating the p-values directly to the figure legend. Parametric or not paramentric? Has the normality of the distribution been tested?
– Figure 2. Why here is reported both a parametric and non-parametric test? Have been the assumption of the two tests verified?
– Page 4, lines 169 - 170. The authors state: “An increased α-diversity was observed at the T1 time-point compared to the T0, regardless of the absence of statistical significance (p-value> 0.05, ANOVA test)”. The increase is not statistically significant. If any, I would define it as a “slight” increase. Then, each a-diversity metric has its role and should be explained. Chao1 is a measure of richness and goes a bit with the observed OTUs. Shannon includes richness and evenness. Did the authors consider evaluating dominance?
– Supplementary Figure S2 legend: “b-diversity assessed by”. What are the graphs representing? PCoA on different B-diversity metrics? All figure legends should be carefully revised and contain all information necessary for interpretation.
– Page 5, lines 176 and on. “The b-diversity, assessed by Bray-Curtis, Euclidian distance, unweighted and weighted UniFrac algorithms, did not provide statistically significant differences between microbial communities at T0 and T1. time points (Figure S2).” How was it tested? PERMANOVA? Please be clear on what you did.
– Lines 183 - 184 “with an increase in Actinobacteria and Bacteroidetes and a decrease in Proteobacteria, respectively, at the T1 time-point (Figure S6) “. These are not statistically significant. It should be clearly stated.
– Can authors use everywhere “phylum”, “family”, or “genus” rather than L2, L5, L6? It would improve readability.
– From line 208 to line 234. This part is not clear at all to me. What are the authors trying to achieve? How is this linked with probiotic treatment? What are the actual results? They should be presented logically and in an understandable way. For example, what are all those KEGG groups? Right now, just a list of IDs. Is there any sense in doing this analysis here? What groups are found? Overall, their function and role should be disclosed in the results, not just added to the “Discussion” section. Further, is there any link between KEGG groups, i.e., functional potential (= it is a prediction by Picrust based on 16s, so there is no assurance you have those functions) and citrulline, IL-12, IL-6, and so on? This actually might be of interest.
– In the discussion, lines 282 - 288. Why do the authors not include these historical data in their work? It would clearly show that the reduction is linked to probiotics supplementation. Otherwise, this part of the discussion is purely speculation.
– The discussion would benefit from a final paragraph summarizing the main findings, not a separate paragraph after the Methods section.
Author Response
Laterza and colleagues analyze the effect of gut microbiota on patients’ recovery from COVID-19 after using a proprietary probiotics formulation. Conflicts of interest are clearly stated. I found the work interesting, but it needs substantial improvements.
>Thank you very much for your consideration of our work
Comments:
I have general considerations on figures (both main figures and supplementary figures). Some are also highlighted explicitly in other points:
– Can the authors overlay single points representing single measurements for all graphs with only summary statistics (boxplot, bar charts, es. Fig 1, 2, S1…)?
>Done, please consider that single points representing single measurements have been reported in the different Figures (Figure 1, Figure 2, Figure S1) where a boxplot is reported. More explanation to the statistics have been provided along the text and the Legends, as requested in the new R1 manuscript version.
– In many figures is not reported the unit of measurement on the y-axis.
>Done, please consider that we have reported for each Figure the unit of measurements on the y-axis in the new R1 manuscript version.
– Figure legends must be exhaustive, indicate what the figure represents, and report information that helps the reader understand.
>Figure 1, 2, 3, 4 have been prepared newly accordingly to requests, while legends for Figure 5, 6, 7 have been improved giving an higher level of details.
– Many figures look like they are copied and pasted straight from the analysis pipelines. A bit of editing would improve the overall perceived quality of the work.
>Done, we have improved the quality of the graphical work. Hopefully, we have reached your requests.
Other issues:
– Page 3, lines 128 - 134. The authors state that some “trends” are emerging. Although it might be true, giving any reasonable scientific evaluation is problematic, considering the small number of enrolled patients and the lack of statistical support. This is also the main issue with this work, as the authors correctly point out.
>We agree with the highlighted point, please see the new paragraph in which we have further stressed the point.
– Figure 1. What is the y-axis representing? The unit of measurement needs to be included.
>Done, please consider the label and new measurements associated to y-axis in the new Figure R1;
-Can the authors improve the readability by showing a point for each measurement (in addition to single box plots) and indicating directly on the graph the significant comparisons?
>Done, please consider the new Figure R1;
-Please also add what statistics you used for calculating the p-values directly to the figure legend. Parametric or not paramentric?
>Done, please consider the new legend of the new Figure R1
-Has the normality of the distribution been tested?
>Done, the normality was tested by qqplot and because the distribution was not normal, we have applied a not parametric test such as Wilcoxon test was exploited to compared paired data.
– Figure 2. Why here is reported both a parametric and non-parametric test?
> We have chosen to report both types of tests to provide information on whether the parametric and non-parametric tests would lead to similar or different conclusions. Indeed, if the results of both tests are consistent, it may provide additional evidence that the assumptions of the parametric test were met. On the other hand, if the results of the two tests differ, it may suggest that the data violate the assumptions of the parametric test or that the non-parametric test is more appropriate for the data.
Therefore, reporting both a parametric and non-parametric test we can provide a more nuanced understanding of the data and the results of our analysis, which can be useful for making informed decisions and drawing appropriate conclusions.
– Have been the assumption of the two tests verified?
>The assumptions for both statistical tests, ANOVA and Kruskal-Wallis, were verified. These assumptions include independence among observations in the dataset, approximately normal distribution of residuals (as confirmed by the Shapiro-Wilk test), homogeneity of variances (as confirmed by Levene's test), and a continuous dependent variable.
– Page 4, lines 169 - 170. The authors state: “An increased α-diversity was observed at the T1 time-point compared to the T0, regardless of the absence of statistical significance (p value> 0.05, ANOVA test)”. The increase is not statistically significant. If any, I would define it as a “slight” increase.
>Done, the text has been amended accordingly, please see the new text.
-Then, each a-diversity metric has its role and should be explained. Chao1 is a measure of richness and goes a bit with the observed OTUs. Shannon includes richness and evenness.
>Done, as requested, the R1 text has been accordingly modified, reporting the specific explanations of the metrices. Chao 1 is an estimator based on abundance; thus, it requires data that refers to the abundance of individual samples belonging to a certain class; Simpson index is the measure of the degree of concentration when individuals are classified into types. follows a similar idea to the Shannon Index it is based on the probability that two entities taken from the sample at random are of different types; Shannon Index is an estimator for both species richness and evenness, but with weight on the richness. The idea behind this metric is that the more taxa you observe, and the more even their abundances are, the higher the entropy, or the higher the uncertainty of predicting which taxa you would see next; Good’s coverage index was used to estimate the percentage of total bacterial ASV represented in a sample; Observed Otus counts up the number of outs you observe; pd whole tree’s phylogenetic diversity measures the amount of the phylogenetic tree covered by the community (Chao, A., & Chiu, C. Species Richness: Estimation and Comparison. Wiley StatsRef: Statistics Reference Online, 2016, 1-26. ).
Did the authors consider evaluating dominance?
>As suggested by the Referee we evaluated also the dominance index. It was calculated using McNaughton’s dominance index included in the R microbiome library (https://rdrr.io/github/microbiome/microbiome/man/dominance.html). Please see the graphical representation of the dominance Index reported as plot in Figure S2.
– Supplementary Figure S2 legend: “b-diversity assessed by”. What are the graphs representing? PCoA on different B-diversity metrics?
> Please see the new Figure legend, reported as follows: A) PCoA plots of sample sets at T0 and T1 time-points based on Bray-Curtis matrices. B) PCoA plots pf T0 and T1 groups based on Euclidean distances matrices. C) PCoA plots of T0 and T1 time-points based on Unweighted Unifrac matrices. D) PCoA plots of T0 and T1 time-points based on Weighted Unifrac. The algorithms did not provide statistically significant differences via the PERMANOVA test (p-value > 0.05) between microbial communities at T0 and T1 time points.
All figure legends should be carefully revised and contain all information necessary for interpretation.
>All information in the Figure legend have been revised accordingly to Reviewer 1 request.
– Page 5, lines 176 and on. “The b-diversity, assessed by Bray-Curtis, Euclidian distance, unweighted and weighted UniFrac algorithms, did not provide statistically significant differences between microbial communities at T0 and T1. time points (Figure S2).” How was it tested? PERMANOVA? Please be clear on what you did.
>Yes, the statistical significance was assessed by permanova. Please see the new legend.
– Lines 183 - 184 “with an increase in Actinobacteria and Bacteroidetes and a decrease in Proteobacteria, respectively, at the T1 time-point (Figure S6) “. These are not statistically significant. It should be clearly stated.
>A sentence in the R1 version of the manuscript to clearly state that these observation are not statistically significative as suggested by the referee has been provided.
– Can authors use everywhere “phylum”, “family”, or “genus” rather than L2, L5, L6? It would improve readability.
>Following referee suggestion, L2, L5, L6 were replaced with “phylum”, “family”, or “genus” respectively.
– From line 208 to line 234. This part is not clear at all to me.
-What are the authors trying to achieve? How is this linked with probiotic treatment?
>Our aim was to identify microbial markers of the gut ecological systems at T0 and T1 before and after treatment of probiotics, by a model of prediction based on machine learning to predict or to classify, just, by the most important features or ASVs, the system at T1 compared to the system at T0. Interestingly, the major point achieved by ML computation was that all the “features” associated to T1 were Bifidobacterium (L6, genus), Bifidobacteriace (L5, family) and Actinobacteria (L2, phylum), consistently with components of the probiotic administered in the study or short chain fatty acids (SCFA) producers, generally linked to probiotic administration.
Moreover, the inferred KOs by PICRUST algorithm were puntcually investigated to verify if any biochemical pathway could be over or under represented at the T0 or T1 time-point. Indeed, by this analysis, a complete disappearence of virulence and antibiotic resistance-related KOs was reported at T1 time point, suggesting an effect directed against cassettes of bacterial virulence and pathogenicity, hence against pathobionts, possibly related to probiotic administration.
-What are the actual results?
- In ML, Bifidobacterium, reported at different taxonomic levels, represent one of the main components of the probiotic administered;
- the inferred KOs identified by PICRUST at T0 (virulence and antibiotic resistance-related KOs) resulted completely lost at the T1 time-point, suggesting a mitigating effect against pathobionts, possibly related precisely to probiotic administration;
- Spearman’s correlation highlighted significant negative correlation between citrulline and ASV at L2 (phylum Actinobacteria), L5 (family Bifidobacteriaceae) and L6 (genus Bifidobacterium), all ASVs referred to Bifidobacterium-linked taxonomic levels, once again a main component of the administered probiotic. In this context, citrulline did not seem to be related to variation on enterocyte mass, but its modification could be related to a shift in GM ecology and bacterial metabolism. In fact, the inverse correlation between serum citrulline and Bifidobacterium spp. probably is due to specific amino acids-linked metabolic pathways, as already reported for GM Akkermansia and not to physiological effects (please see the text).
-The results should be presented logically and in an understandable way.
We have completely rewritten the sentences from line 208 to line 234 of the first manuscript version. Please see the new paragraph in the R1 version of the manoscript regarding ML, PICRUST and Spearman’ correlation results at page 22.
-For example, what are all those KEGG groups? Right now, just a list of IDs. Is there any sense in doing this analysis here? What groups are found? Overall, their function and role should be disclosed in the results, not just added to the “Discussion” section.
>Please consider in this R1 version of the Results the most detailed explanation of the meaning of the KOs linked to T0 and T1 ecological gut microbiota systems. In particular, functional microbial KOs, such as microbial virulence factors and antibiotic resistance operon were described and then discussed as only T0-related KOs, therefore unique.
-Further, is there any link between KEGG groups, i.e., functional potential (= it is a prediction by Picrust based on 16s, so there is no assurance you have those functions) and citrulline, IL-12, IL-6, and so on? This actually might be of interest.
>In correlation heatmaps, Spearman’s correlation was used to examine the association between features (e.g., biochemical markers and ASVs) and only statistically significant correlations (FDR adjusted p values <0.05) were reported. The Spearman’s correlation highlighted significant negative correlation between citrulline and ASV at L2 (Actinobacteria), L5 (Bifidobacteriaceae) and L6 (Bifidobacterium) (FDR adjusted p values <0.05). Also, significant negative correlation between IL6 and ASV was observed but only at L2 (Bacteroidetes) and L5 (Methanobacteriaceae) (p values <0.05).
However, please consider that the correlation between KOs groups and citrulline, IL-12, IL-6 did not provide any statistically significant correlations (data not shown).
– In the discussion, lines 282 - 288. Why do the authors not include these historical data in their work? It would clearly show that the reduction is linked to probiotics supplementation. Otherwise, this part of the discussion is purely speculation.
>Thank you for your suggestion. We have decided to remove this part from the discussion.
– The discussion would benefit from a final paragraph summarizing the main findings, not a separate paragraph after the Methods section.
>Please consider the new organization of the text, in which the conclusion are just adjacent to the discussion

Reviewer 2 Report
The title of this article is “Ecology and machine learning-based classification models of gut microbiota and inflammatory markers may evaluate the effects of probiotic supplementation in patients recently recovered from COVID-19”. This is an interesting topic, and it is an area that needs our attention. However, there are still some areas of the article that need to be revised:
1. Today's coronaviruses have undergone multiple variants, and the authors need to consider whether the current study is still instructive when placed in the present day.
2. Figure 2. In this section, the authors need to explore the results in more depth and check the results obtained with recently published journals.
3. The "Materials and Methods" section of the article. The author needs to further organize this section, such as adding some new subheadings or merging some similar content to make the article more concise.
4. The authors may consider improving the layout of the article, for example, placing Chapter 4 "Materials and Methods" in the second chapter of the article, so that the reader can better understand the experimental operation of the article.
5. The article mentions the association of intestinal flora with COVID-19 after probiotic supplementation. For this section, the authors need to indicate the strains of bacteria that appear significantly altered based on the changes in the structure of the intestinal flora that occur after probiotic supplementation and discuss in depth what benefits this alteration would provide to the patient.
6. Authors are requested to carefully check the format of the references used in the article to ensure that the references are in the required format.
Author Response
The title of this article is “Ecology and machine learning-based classification models of gut microbiota and inflammatory markers may evaluate the effects of probiotic supplementation in patients recently recovered from COVID-19”. This is an interesting topic, and it is an area that needs our attention. However, there are still some areas of the article that need to be revised:
1.Today's coronaviruses have undergone multiple variants, and the authors need to consider whether the current study is still instructive when placed in the present day.
>Thank you for this suggestion. Please see our comment in the discussion, as follow: “Since the first outbreak in 2019, SARS-CoV-2 has undergone multiple variants over time. These variants have developed mutations capable of conferring higher transmissibility or antigenicity. However the effect of probiotic supplementation demonstrated in this paper is not expected to change with different SARS-CoV-2 variants, as it is mostly related to probiotics contained in the mix rather than to the interaction with the virus”.
2.Figure 2. In this section, the authors need to explore the results in more depth and check the results obtained with recently published journals.
> Thank you for this comment. Please consider that new paragraph reported in the new Result Section of the R1 manuscript version regarding the explanation of the a-diversity, with special reference to Figure 2 that has been completely rewritten.
To date, in our knowledge no studies have been reported relationships between probiotic administration and diversity values in long COVID-patients in term of ecology dynamics.
3.The "Materials and Methods" section of the article. The author needs to further organize this section, such as adding some new subheadings or merging some similar content to make the article more concise.
>Accordingly, the Section 4.1 (Study design), 4.1.1 (Study aims), 4.1.2. (Study treatment), 4.1.3. (Study population), can be merged into the single Section 4.1 Study design and aims. Moreover, the sections 4.1.4. (Outcome measures), 4.1.4.1. (Measures of clinical outcomes and treatment satisfaction) can be merged into the Section 4.2 Measures of clinical outcomes. The sections 4.2. (Measures of immunological response and intestinal barrier integrity), 4.3. (Measures of intestinal and systemic inflammation) can be merged into the Section 4.3 Measures of laboratory outcomes. After these section has ben restablished the Section 4.4 (Statistical analysis of metadata). Please see the below reported request 4
4.The authors may consider improving the layout of the article, for example, placing Chapter 4 "Materials and Methods" in the second chapter of the article, so that the reader can better understand the experimental operation of the article.
>Accordingly, the Chapter 4 has been moved into the chapter 2 of the article. However, please consider that the template for IJMS manuscript request Methods at the end of the manuscript. Moreover, the Sections have reported as requested at the point 3 of the current revision
5.The article mentions the association of intestinal flora with COVID-19 after probiotic supplementation. For this section, the authors need to indicate the strains of bacteria that appear significantly altered based on the changes in the structure of the intestinal flora that occur after probiotic supplementation and discuss in depth what benefits this alteration would provide to the patient.
>Due to the technical limitations of 16S rRNA gene sequencing, it is generally not possible to reach the strain level of taxonomic resolution. Additionally, the sequencing reads generated by 16S rRNA gene sequencing are typically short (around 150-250 base pairs), which makes it difficult to resolve differences between closely related bacterial strains (Jovel, J., Patterson, J., Wang, W., Hotte, N., O'Keefe, S., Mitchel, T., Perry, T., Kao, D., Mason, A.L., Madsen, K.L., and Wong, G.K. (2016). Characterization of the gut microbiome using 16S or shotgun metagenomics. Frontiers in Microbiology, 7: 459). However a persistent association of the entire ecology with the prrsence of bacteria at Bifidobacterium genus level is highlighted by different biocomputing algorithms.
6.Authors are requested to carefully check the format of the references used in the article to ensure that the references are in the required format.
>Done, the references have been completely checked in term of required format

Reviewer 3 Report
This manuscript can be accepted after a few language revisions.
Author Response
This manuscript can be accepted after a few language revisions.
>Thank you very much for the appreciation of our work. A careful revision of the language has been performed by a mother tongue person.

Round 2
Reviewer 1 Report
All the comments were addressed in a proper way. The current version has been really improved. Congratulations to the authors for the excellent work.